# Immunomodulatory Properties of Natural Extracts and Compounds Derived from *Bidens pilosa* L.: Literature Review

**DOI:** 10.3390/pharmaceutics15051491

**Published:** 2023-05-13

**Authors:** Xandy Melissa Rodríguez-Mesa, Leonardo Andres Contreras Bolaños, Antonio Mejía, Luis Miguel Pombo, Geison Modesti Costa, Sandra Paola Santander González

**Affiliations:** 1Phytoimmunomodulation Research Group, Juan N. Corpas University Foundation, Bogotá Carrera 111 #159A-61, Bogota 111321, Colombia; 2Plant Pharmacology and Alternative Therapeutics, Juan N. Corpas University Foundation, Bogotá Carrera 111 #159A-61, Bogota 111321, Colombia; 3Phytochemistry Research Group (GIFUJ), Pontificia Universidad Javeriana, Bogotá Carrera 7 #40-62, Bogota 110231, Colombia

**Keywords:** *Bidens pilosa* L., immunomodulation, phytochemistry, traditional medicine, phytotherapy

## Abstract

*Bidens pilosa* L. has been used in different parts of the world mainly to treat diseases associated with immune response disorders, such as autoimmunity, cancer, allergies, and infectious diseases. The medicinal properties of this plant are attributed to its chemical components. Nevertheless, there is little conclusive evidence that describes the immunomodulatory activity of this plant. In this review, a systematic search was carried out in the PubMed-NLM, EBSCO Host and BVS databases focused on the pre-clinical scientific evidence of the immunomodulatory properties of *B. pilosa*. A total of 314 articles were found and only 23 were selected. The results show that the compounds or extracts of *Bidens* modulate the immune cells. This activity was associated with the presence of phenolic compounds and flavonoids that control proliferation, oxidative stress, phagocytosis, and the production of cytokines of different cells. Most of the scientific information analyzed in this paper supports the potential use of *B. pilosa* mainly as an anti-inflammatory, antioxidant, antitumoral, antidiabetic, and antimicrobial immune response modulator. It is necessary that this biological activity be corroborated through the design of specialized clinical trials that demonstrate the effectiveness in the treatment of autoimmune diseases, chronic inflammation, and infectious diseases. Until now there has only been one clinical trial in phase I and II associated with the anti-inflammatory activity of *Bidens* in mucositis.

## 1. Introduction

*Bidens pilosa* L. is a plant belonging to the Asteraceae family; the genus Bidens (Figure 1) includes about 280 species. It is commonly known as: chipaca, amor seco, masaquía sillcao, cadillo, picao preto, cuamba, Spanish needles, beggar’s ticks, devil’s needles, cobbler’s pegs, broom stick, pitchforks, and farmers’ friends [1,2,3,4,5,6]. Although considered native to South America, B. pilosa is distributed in pantropical areas, such as Colombia, Brazil, Peru, Uganda, Kenya, China, Australia, and Hawaii [7]. The traditional use of this plant has been recorded throughout the Americas, Africa, Asia, and Oceania [2,3,4,5,6,7,8].

Furthermore, its medicinal properties have been used in treating more than 40 illnesses, such as asthma, pharyngitis, diabetes, gastritis, infectious diseases, cancer, and also for the management of wounds and inflammation [2,9,10,11]. This is mainly due its diverse chemical composition.

Approximately 201 metabolites have been described, such as flavonoids, terpenes, phenylpropanoids, aromatic, aliphatic compounds, and porphyrins [2,3,4,5]. In the group of flavonoids and polyynes, such as centaureidin, centaurein, luteolin, and cytopiloin (Cp), it has been documented that these have antiallergic, antioxidant, antiproliferative, immunosuppressive, and anti-inflammatory properties [2,5,7,12,13,14].

Many current treatments for chronic inflammatory diseases, such as autoimmune conditions and cancer, cause unwanted side effects and, in many cases, are ineffective [15,16,17]. Therefore, in recent years, the research focused on developing therapeutic strategies based on immunomodulation has gained great importance [18,19,20], especially those inspired by the use of products of natural origin that can modulate the increase or decrease in the immune response.
Figure 1Watercolors of *B. pilosa* L. Creator: Frances Worth Horne. Taken from: Archives of The New York Botanical Garden [21].
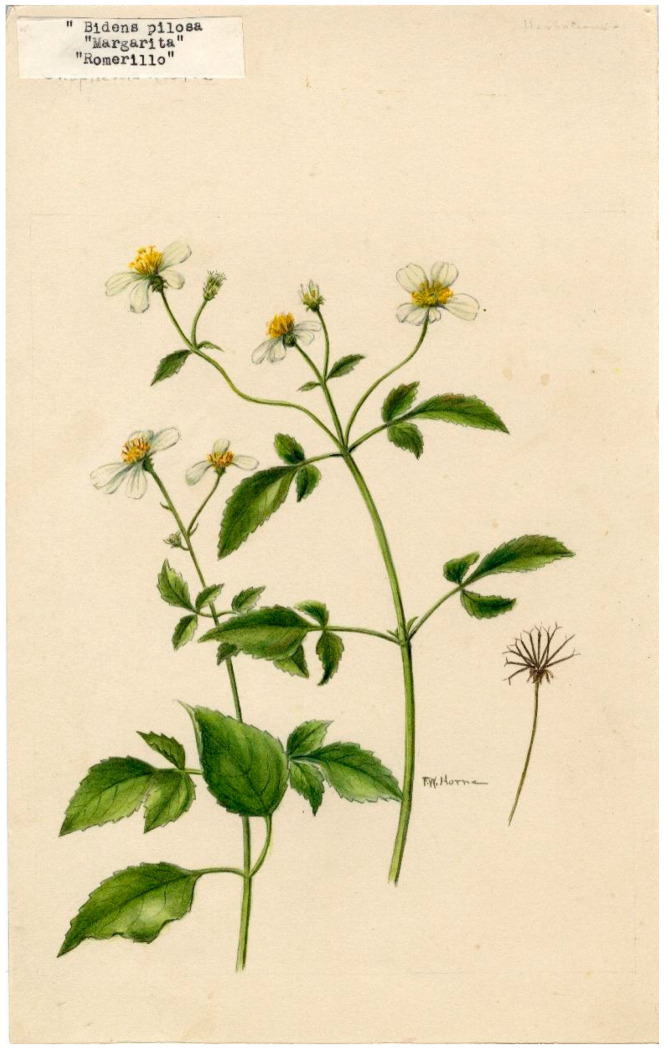



## 2. Materials and Methods

The literature search was conducted in the National Library of Medicine (PubMed-NLM), EBSCO Host, and Virtual Health Library (VHL) databases, using different descriptors in health sciences (DeCS) and medical subject headings (MeSH) terms previously determined as “Bidens pilosa” or “Bidens” combined with the terms described in Table 1.

A total of 1078 publications were found. During the selection process, publications about experimental models, published between 2010 and 2020 and that were in English were included. With these criteria, 547 results were obtained. Duplicate articles were discarded using the Zotero software, yielding 314 individual articles. Titles and abstracts were then screened. Studies that did not discuss the obtaining of extracts or compounds from the plant or its chemical composition or biological effects related to activity on the immune system in different experimental models were excluded. In this way, 20 articles were selected, but 3 articles published prior to 2010 were added due to the relevance of the scientific evidence they reported and the frequency of references to them in recent works. Finally, 23 articles were included for the review.

## 3. Results

### 3.1. B. pilosa Extracts Have Anti-Inflammatory Properties Associated with Antioxidant and/or Analgesic Effects

The immune system response is unbalanced in some diseases, a clear example being autoimmune conditions, chronic inflammation, cancer, chronic kidney disease, cirrhosis, and neurodegenerative disorders. These diseases have a high prevalence, cause disability, and have a high mortality rate worldwide [22,23,24]. Although there are different medicines for managing these conditions, different medicinal plants are currently under investigation in search of new phytomedicines or polymolecular medications for the management of symptoms or regulation of the immune response with fewer adverse effects.

In traditional medicine, *B. pilosa* has been used mainly to treat different inflammation-associated symptoms and diseases. However, scientific evidence supports its traditional use and suggests new effects associated with its anti-inflammatory properties (Table 2).

An in vivo study evaluated the analgesic and anti-inflammatory properties of the ethyl acetate fraction (EtOAc) of an extract obtained from the leaves of *B. pilosa* in models of chemical and thermal nociception in mice and Wistar rats [25]. Oral administration of this fraction at doses of 50, 100, and 200 mg/kg showed a significant antinociceptive effect in four different models. EtOAc inhibited the number of contortions recorded during a period of 15 minutes (min) induced by acetic acid and delayed the reaction time on the hot plate model in a dose-dependent manner to 12.40 ± 0.36 seconds (s), 18.40 ± 0.22 s, and 20.80 ± 0.33 s, respectively. The time spent licking the paw following capsaicin injection was significantly and dose dependently reduced by EtOAc at 54.00 ± 0.55 s, 49.80 ± 0.58 s, and 45.40 ± 0.51 s, respectively. In the carrageenan- and dextran-induced visible paw edema model, the treatment with *B. pilosa* at 200 mg/kg antagonized the formation of edema. A fraction had a similar effect to the experimental control of acetylsalicylic acid. Therefore, *B. pilosa* could be inhibiting the neurogenic pain via peripheral mechanisms in a similar way to acetylsalicylic acid through the inhibition of cyclooxygenase (COX) enzyme activity and the inhibition of the release of other endogenous pain mediators. In addition, the reduced inflammation and oedema observed were probably associated with the flavonoids, quercetin 3,3′-dimethyl ether 7-*O-β-D*-glucopyranoside (Quercetin) and iso-okanin 7-*O-β-D*-glycopyranoside identified in this fraction; these compounds have been recognized for their anti-inflammatory and antinociceptive properties [25].

Quercetin is an important bioflavonoid found in more than twenty plants. Its anti-inflammatory, antioxidant, antihypertensive, vasodilator, anti-obesity, anti-hypercholesterolemic, and anti-atherosclerotic effects are well-known [26]. The presence of this flavonoid in *B. pilosa* confirms the anti-inflammatory properties reported in the plant and suggests new pharmacological uses associated with its quercetin content.

In another in vivo study, De Ávila et al. evaluated the effect of a glycolic extract of *B. pilosa* (Ecobidens^®^) in a poloxamer-based mucoadhesive formulation, in a murine model of 5-fluorouracil (5-FU)-induced intestinal mucositis [27]. 5-FU is a chemotherapeutic commonly used for the treatment of cancer. Its use induces a reduction in intestinal cell proliferation associated with an increasing rate of apoptosis, which in turn leads to a loss of intestinal arrangement and function, mainly through villus shortening and crypt destruction. Oral administration of the *B. pilosa* formulation for 6 days, mainly at a dose of 100 mg/kg, restored the proliferative activity of intestinal cells as evidenced by a high level of the Ki-67 antigen (Ki-67) (a marker of increased cell proliferation) in the intestinal villi and crypts. Additionally, the *B. pilosa* formulation also downregulated the expression of apoptotic markers, such as Bcl-2-like protein 4 (Bax) and phosphoprotein p53 (p53) in intestinal cells, protecting them from death. Likewise, the plant extract regulated lipid peroxidation and inflammatory infiltration in the intestinal tissue by reducing the activity of the myeloperoxidase (MPO), an enzyme released by neutrophils during the inflammatory process, which favors cellular stress and the production of malondialdehyde (MDA), the latter being a stress marker in intestinal mucositis [27].

This research group subsequently reported similar findings when a mucoadhesive formulation containing Ecobidens^®^ and curcuminoids from *Curcuma longa* L. was evaluated in the same model of intestinal inflammation [28], in search of an effective treatment for the intestinal mucositis caused as a side effect of chemotherapy in patients with cancer. Considering that both plants exert separate anti-inflammatory and immunomodulatory effects, the authors tested the efficacy of a combined plant formulation. The orally administered formulation (125 mg/kg of *B. pilosa* + 15 mg/kg of *Curcuminoids*) for 6 days showed a protective effect on the intestinal epithelium when exerting anti-inflammatory and antioxidant activities. These effects were associated with an increased expression of Ki-67 and Bcl-2 proteins in intestinal villi and crypts, in addition to increased IL-10 levels, reduced Bax expression, and decreased MPO and MDA activities in the intestinal tissue. These results indicated that the combined mucoadhesive formulation had a remarkable capacity to downregulate the apoptosis rate and restore the proliferative activity of the intestinal cells, and effectively reduce the 5-FU-induced intestinal injury [28].

This combined plant formulation, subsequently named FITOPROT, was also evaluated by the same group in an in vitro study on human keratinocyte cell line (HaCaT) exposed to 5-FU [29] during 24 hours (h). Similar to the results observed in animal models, FITOPROT showed antioxidant and anti-inflammatory effects by significantly reducing the production of reactive oxygen species (ROS). In addition, it avoided mitochondrial changes by preventing the release of mitochondrial cytochrome C to the cytoplasm and restored the cell proliferative activity, as evidenced by the Ki-67 expression. In addition, the plant formulation regulated the 5-FU-induced oxidative stress through a mechanism involving the nuclear factor erythroid 2-related factor 2 (Nrf2). Additionally, HaCaT cells pre-treated or treated with FITOPROT showed normal expression of inflammation-related proteins, such as tumor necrosis factor receptor 1 (TNFR1) and the nuclear factor-κB (NF-κB), and reduced levels of pro-inflammatory cytokines, such as tumor necrosis factor (TNF), interleukin-1β (IL-1β), IL-6, and IL-8 [29].

The combination of active metabolites of medicinal plants is of great importance because additive or synergistic interactions can enhance their biological activity in inflammatory diseases. It is also essential to note that many of these diseases are not regulated by a single molecular target but often have a multifactorial causality [30,31]. Resistance to treatment is less likely to occur when active compounds derived from different plants are used in combination than when they are used individually [32,33], because they can act on multiple molecular targets in different cells of the immune system associated with the development of diseases. Considering the results obtained with FITOPROT, it would be interesting to evaluate whether its therapeutic scope could be extended to other illnesses with similar inflammatory issues, such as Crohn’s disease and ulcerative colitis, whose progression is accompanied by chronic intestinal inflammation and relates to an exacerbated production of Th1 and/or Th2 cytokines [34,35].

Another *B. pilosa* extract that has been studied for its anti-inflammatory properties is the aqueous extract obtained from the infusion of the whole plant, excluding the root. This extract was tested in a model of Wistar rats treated with carbon tetrachloride (CCl_4_). Degradation of CCl_4_ generates unstable free radicals such as trichloromethyl (CCl_3_), which induces depletion of the antioxidant status, deoxyribonucleic acid (DNA) damage, liver inflammation, kidney damage, and damage to other organs, such as the heart, brain, testicles, and lungs, due to excessive production of free radicals [36,37,38]. CCl_4_ was administered intraperitoneally (1 mL of solution per kilogram of body weight) twice a week for 10 weeks, and the treatment with the aqueous extract was administered orally by gavage in 0.5 mL of *B. pilosa*/100 g daily and/or topically by daily bathing in the tea. Animals were trichotomized with a trichotomy device throughout the back (cervical to lumbar regions) and abdomen for better absorption of the extract. *B. pilosa* effectively protected against the toxic effects of CCl4, due to the biochemical parameters becoming normalized, there was no hepatic inflammation, and the degree of inflammation in the intestinal mucosa decreased in the animals. Combined use (oral and topical) exerted a better protective effect [39]. Another study also described the anti-inflammatory effects of the aqueous extract obtained from the *B. pilosa* leaves in a rat model of androgen deficiency-related dry eye disease. This condition is characterized by unstable tear production, inflammation of the ocular surface epithelium and lacrimal glands, and secretory dysfunction of the glands lacrimal and secretory dysfunction in which pro-inflammatory signaling pathways, such as mitogen-activated protein kinase (MAPK) and NF-κB, are activated, favoring the pathophysiology [40,41]. This study showed that ocular inflammation resulted from an imbalance between inflammatory and anti-inflammatory responses. The aqueous extract of *B. pilosa*, orally administered once a day for 4 weeks, significantly decreased the expression of pro-inflammatory mediators, such as IL-1β, Fas ligand (FasL), and TNF-α in the long-term and reduced lymphocyte infiltration, alleviating lacrimal gland inflammation, and improving tear secretion and tear film stability [42].

It will be essential to characterize the most abundant compounds in the aqueous extracts of *B. pilosa* and elucidate their anti-inflammatory properties. This characteristic is usually attributed to apolar secondary metabolites rather than polar primary metabolites, most likely present in plant aqueous extracts. No studies on the immunomodulatory activity of primary metabolites of *B. pilosa* were found in the literature reviewed for the present work. Therefore, further investigations are needed to establish which polar metabolites might be associated with the anti-inflammatory activity reported in the inflammation models mentioned above.

*B. pilosa* is a plant enriched in fatty acids, which have been associated with anti-inflammatory effects through the modulation of oxidative stress and the gene expression of inflammatory mediators. An investigation tested the anti-inflammatory properties of a fatty acid-enriched supercritical fluid (SCF) from *B. pilosa* in the model of TBNS-induced inflammatory bowel disease in rats and Swiss mice [43]. This extract, orally administered at 25, 50, or 100 mg/kg at 96, 72, 48, 24, and 2 h before intestinal inflammation induction, showed anti-inflammatory effects associated with increased IL-10 synthesis, reduced oxidative stress, and reduction in the pro-inflammatory cytokines IL-1β, IL-6, and TNF-α. Furthermore, the beneficial effects of *B. pilosa* were closely related to the downregulation of heparanase signaling, 70-kDa heat shock proteins (Hsp70), MAPKs, NF-κB, and a significant increase in mucin 3 and 4 gene expression [43]. These results show that SCF has a promising preventive effect, through the regulation of these pathways associated with the progression of inflammation, favoring the integrity of the epithelium and intestinal mucosa.

According to this study, results from our research group (article in preparation) showed that fatty acids and terpenes would be involved in the decreased production of inflammatory cytokines by human peripheral mononuclear cells (PBMCs) and in the polarization of human macrophages to the anti-inflammatory profile.

In 2020, Abiodun et al. reported the effect of *B. pilosa* in the model of 2,4,6 trinitrobenzene sulfonic acid (TNBS)-induced colitis in Wistar rats. They found that a methanolic extract obtained from the whole plant, administered orally for 9 days (2 days pretreatment prior to colitis induction, followed by 7 days of treatment post-colitis induction) at 200 and 400 mg/kg, significantly reduced inflammation-associated colon injury. This anti-inflammatory effect would be closely related to the antioxidant activity exerted by the components of the methanolic extract since the results showed a reduced level of lipid peroxidation in the colon. It is known that lipid peroxidation alters the integrity of the intestinal mucosal barrier and activates the synthesis of proinflammatory cytokines, such as IL-6 and IL-8 [44,45].

A study by Pereira et al. evaluated the activity of a methanolic extract of *B. pilosa* leaves and of a polyacetylene compound (2-*O-β-D*-glucosyltrideca-11*E*-en-3,5,7,9-tetrayn-1,2-diol) isolated from this extract in two in vitro models: human PBMCs stimulated by phytohemagglutinin (PHA) or by 12-*O*-Tetradecanoylphorbol-13-acetate (TPA) plus ionomycin, and murine lymphocytes stimulated by concanavalin A (ConA). The extract was tested in an in vivo model of murine zymosan-induced arthritis, which was administered intraperitoneally at a concentration of 10 mg extract from day 2 to day 6 after zymosan injection. The extract and the isolated compound showed anti-inflammatory effects and antiproliferative activity. Although the extract inhibited the proliferative response in the in vitro models, the isolated compound was 10-fold more potent for blocking proliferation (IC_50_ = 1.25 to 2.5 µg/mL). Additionally, a correlation was found between the antiproliferative activity of the extract and the anti-inflammatory effects observed in the mouse model [10].

Interestingly, these studies demonstrated that the immunomodulatory properties of the methanolic extract of *B. pilosa* were not limited only to the intestinal inflammation model, as they appeared to have a potential effect on other autoimmune diseases such as arthritis. However, further preclinical studies with polyacetylenes are needed to determine their activity in these diseases.
pharmaceutics-15-01491-t002_Table 2Table 2Scientific evidence on the anti-inflammatory, antioxidant, and analgesic effects of *B. pilosa*.Part of the PlantExtract TypeExperimental Model TypeBiological EffectMetabolites Isolatedor Identified Study ConclusionsReferenceLeavesEthyl acetate fraction Chemical and thermal models of nociception in *Mus musculus* mice and Wistar ratsIn vivoAnti-inflammatory and analgesicQuercetin 3,3′—dimethyl ether 7-*O-β-D*-glucopyranosideIso-okanin 7-*O-β-D*-(2″,4″,6″-triacetyl)- glycopyranosideNeurogenic pain inhibitionPain relief through peripheral mechanismsOedema inhibition[25]Not specifiedEcobidens^®^(Glycolic extract in a poloxamer-based liquid formulation)5-FU-induced intestinal mucositis in male Swiss miceIn vivoAnti-inflammatory, antioxidant, and intestinal protector-Restoration of proliferative activity of intestinal cells↑ Ki-67 levelsModulation of the expression of Bax and p53Regulation of lipid peroxidation and inflammatory infiltration↓ MPO y MDA[27]Not specifiedEcobidens^®^) in mucoadhesive formulation with curcuminoids5-FU-induced intestinal mucositis in male Swiss miceIn vivoAnti-inflammatory, antioxidant, and intestinal protector-↑ Ki-67 and Bcl-2 expression↓ Pro-apoptotic regulator Bax, activity of MPO and MDA↑ IL-10 levels[28]Not specifiedFITOPROT (Formulation of glycolytic extract *B. pilosa*, poloxamer, and curcuminoids)HaCaT cell line exposed to 5-FUIn vitroAnti-inflammatory and antioxidant-↓ ROS, membrane potential, and cytochrome C releaseMaintenance of cell proliferative capacity↓ Oxidative stress due to Nrf2 involvement Normal expression of TNF-R1 and NF-κB↓ Production of TNF, IL1β, IL-6, and IL-8[29]Whole except rootAqueous (infusion)Wistar rats treated with CCl_4_In vivoAnti-inflammatory-Protective effect on liver, intestine, and kidney injury↓ Hepatic and intestinal inflammationRenal tubular regeneration↓ Intestinal mucosa inflammation in exposed animals[39]Dry leavesAqueous extractDry eye model in healthy female Wistar ratsIn vivoAnti-inflammatory-↓ IL-1β, FasL and TNF-α.↓ Leukocyte infiltration and inflammatory response in the lacrimal glands[42]AerialSCFTNBS-induced intestinal inflammation in male Wistar rats and male Swiss miceIn vivoAnti-inflammatory and antioxidantFatty acids↑ IL-10 production↓ Level of IL-1β, IL-6, and TNF-α↓ Oxidative stress↑ Mucin production↓ Heparanase, Hsp70, MAPK3, and NF-κB signalingBlocking of MAPK signaling pathways [43]WholeMethanolic extractTNBS-induced colitis in Wistar ratsIn vivoAnti-inflammatory and antioxidant-↓ Leukocytes’ infiltration and TNF-α productionInhibition of oxidative stress and pro-inflammatory cytokines[44]Dry leavesMethanolic extractHuman PBMCs, murine lymphocytes, and B10.ArSg SnJ mice with Zymosan-induced arthritisIn vitro/vivoAntiproliferative, and anti- inflammatoryPolyacetylene2-*O-β-D*-glucosyltrideca-11E-en-3,5,7,9-tetrayn-1,2-diol↓ Proliferation of human lymphocytes (Extract)↓ Peripheral node swelling at the site of injury induced by zymosan↓ Murine lymphocyte proliferation (Isolated Compound)[10]Abbreviations: (5-FU) Fluorouracil 5-FU; (Bax) BCL-2-like protein 4; (CCl_4_) Carbon tetrachloride; (FasL) Fas ligand; (HaCaT) Human epidermal keratinocyte; (Hsp70) 70-kDa heat shock protein; (IL) Interleukin; (Ki-67) Ki-67 antigen; (MAPK) Mitogen-activated protein kinase; (MDA) Malondialdehyde; (MPO) Myeloperoxidase; (NF-κB) Nuclear factor-κB; (Nrf2) Nuclear factor erythroid 2-related factor 2; (p53) Phosphoprotein p53; (PBMCs) Peripheral blood mononuclear cells; (ROS) Reactive oxygen species; (SCF) supercritical fluid; (TNBS) 2,4,6 trinitrobenzene sulfonic acid; (TNF) Tumor necrosis factor; (TNFR1) Tumor necrosis factor receptor 1; ↑ Increase; ↓ Decrease.


### 3.2. The Methanolic Extract, the Essential Oil, and the Cp Obtained from B. pilosa Modulate the Activation of the Immune System and Induce a Protective Effect against Infectious Diseases

Although the discovery of antimicrobials has been one of the most outstanding milestones in modern medicine, the rise of antimicrobial resistance has generated a public health problem [46]. For this reason, multiple investigations have focused on the search for new immunomodulatory medicines that can trigger the immune response to generate or induce host protection against microorganisms.

The studies summarized in Table 3 show some extracts and compounds obtained from *B. pilosa* with potential antimicrobial properties; in some cases, they have been able to modulate the immune response in the context of infectious diseases.

One of the components of *B. pilosa* that has been widely studied is Cp. This polyacetylenic constituent is usually obtained in the methanolic extract of the whole plant and has an antimicrobial effect against *L. monocytogenes* in a model in vivo. Mice received an intraperitoneal injection of Cp (1.5, 3.125, 6.25, 12.5, and 25 μg/kg) three times per week for 2 weeks. After 24 h, mice were intraperitoneally injected with *Listeria*. Cp increased the survival rate in a dose-dependent manner. Likewise, mice’s liver and spleen showed lowered counts of bacterial colony-forming units (CFU) and less severe lesions [47].

These authors also reported that in an in vitro model of *Candida parapsilosis* infection, Cp enhanced the protein kinase C (PKC)-dependent phagocytic activity and induced the intracellular death of yeast through phagosomal acidification and lysosomal enzymatic activity of RAW264.7 macrophages. In a murine model, mice received an intraperitoneal injection of Cp (5, 12.5, and 25 μg/kg) three times per week for 2 weeks. The treatment decreased the spread of *Candida* dissemination, and reduced hepatic and splenic lesions in infected mice [48].

On the other hand, resistance to antiparasitic agents is also raising concerns in the poultry industry, especially in the case of avian coccidiosis. This infection causes a high mortality rate, reduced growth, and increased treatment costs [49,50,51,52]. An alternative strategy to conventional anticoccidial agents is the use of plant-derived compounds [52]. Lohmann female chicks were fed daily during 21 days with a diet containing *B. pilosa* powder at the dose of 0.05% (0.5 g *B. pilosa*/kg diet), 0.01% (0.1 g *B. pilosa*/kg diet) or 0.002% (0.02 g *B. pilosa*/kg diet), or with Cp at 500 ppb, 100 ppb, or 20 ppb. *B. pilosa* significantly suppressed *E. tenella* infection as evidenced by the reduction in mortality rate, oocyst excretion, and gut pathological severity in chickens. Cp directly interfered with the intracellular life cycle of the protozoan, as demonstrated in Madin-Darby bovine kidney cell (MDBK) cultures. Additionally, Cp reduced the occurrence of periocular dehydration and cecal hemorrhage in a dose-dependent manner and decreased the mortality of infected chickens [52]. This work also demonstrated that a methanolic extract of *B. pilosa* enhanced immunity through the increased expression of interferon-gamma (IFN-γ) in T cells isolated from chicken cecal tonsils, which are important components of host immunity against coccidiosis. However, further studies are required to evaluate whether Cp is responsible for this effect [52,53].

Although these in vivo studies of Cp are noteworthy, further research is required to determine whether this component would have the same immunomodulatory effect on human cells against pathogens such as *Listeria* and *Candida*. Future research will also establish whether Cp or other polyacetylenic components might have a protective role against public health-relevant human infections caused by *Cryptosporidium* spp., *Cyclospora* spp., or *Cystoisospora* spp.

An in vitro study carried out by Liu et al. evaluated the antibacterial effect of several monofloral honeys produced by *Apis mellifera* from the nectar of *B. pilosa*, *Dimocarpus longan*, *Litchi chinensis*, *Citrus maxima*, and *Aglaia formosana*. The disk-diffusion method showed that the sensitivity to each honey sample differed among bacteria tested. However, all honeys exhibited antimicrobial activity against Gram-positive (*S. aureus*, *S. intermedius B*, *S. xylosus*) and Gram-negative (*C. koseri*, hemolytic *E. coli*, and *S. choleraesuis*) bacteria; the diameters of the respective growth inhibition zones in the presence of honey from *B. pilosa* were 3.37 ± 0.04, 3.80 ± 0.13, 3.90 ± 0.07, 2.70 ± 0.13, 3.17 ± 0.04, and 2.50 ± 0.07 [54].

Similarly, Shandukani et al. compared the antimicrobial activity of extracts obtained with hexane, dichloromethane, ethyl acetate, acetone, and methanol of *B. pilosa* and *Dichrostachys cinerea*. They observed that the phenolic compounds of *B. pilosa* had antimicrobial activity against bacteria associated with diarrhea. The minimum inhibitory concentration (MIC), defined as the lowest concentration of the crude plant extract that inhibits bacterial growth after the incubation period, showed that the antibacterial activity of the medicinal plant extracts varied according to the bacteria tested and the type of solvent used for the extraction. For example, the dichloromethane extracts of both plant species displayed high antibacterial activity against all the bacteria tested (*Klebsiella pneumoniae*, commercial probiotics, *E. coli*, *Salmonella typhimurium*, *Shigella boydii*, and *Vibrio parahaemolyticus*) with an average MIC of 0.56 mg/mL. The lower antibacterial effects, particularly in the case of *B. pilosa*, were related to non-polar solvents, which also agree with a predominant antioxidant activity and a high phenolic content [55].

On the other hand, a report described that the essential oil of *B. pilosa* made the bacterial cell membrane more permeable upon interacting with its lipids. The oil caused the death of clinically relevant periodontopathic bacteria, such as *A. actinomycetemcomitans* and *P. gingivalis*, as evidenced by the bacterial growth inhibition determined by CFU counting [56,57].

Since new therapies for periodontitis are based on active compounds that reduce the microbial biofilm and modulate inflammation to reduce tissue destruction, a subsequent study evaluated some vegetable essential oils (*Ocimum gratissimum*, *Cymbopogon nardus*, *Zanthoxylum chalybeum*, and *B. pilosa*) on human gingival fibroblasts [58,59,60]. IL-1β-pretreated fibroblast, exposed to oils produced different levels of prostaglandin E2 (PGE2). Interestingly, *B. pilosa* (20 μg/mL) and *C. nardus* (30 μg/mL) oils enhanced the PGE2 synthesis, whereas the *O. gratissimum* oil diminished it. The effects of *Bidens* and *C. nardus* oils could be associated with their content of phenolic compounds, aromatic amines, and other antioxidant components [58,59,60,61].

It is important to highlight two interesting aspects of *B. pilosa*-derived metabolites; first, they modulate the antimicrobial immune response and second, they have direct antimicrobial activity on microorganisms. The combination of both features broadens the therapeutic spectrum of *B. pilosa* for treating infectious diseases.

Other aspects of *B. pilosa* phytocompounds that deserve further evaluation are the synergy with conventional antimicrobial agents and the capacity to counteract antimicrobial resistance mechanisms, as has been described for other natural compounds derived from *Hydrastis canadensis* L., *Berberine*, and *Vitellaria paradoxa* [62,63,64,65,66,67].

A comprehensive understanding of all these aspects of *B. pilosa* would provide an overview of the multiple targets of extracts or complex mixtures derived from the plant. In addition, it would further support the need for clinical studies to promote the design of effective phytomedicines to treat infectious diseases.
pharmaceutics-15-01491-t003_Table 3Table 3Scientific evidence on the immunomodulatory properties of *B. pilosa* L. that favor an antimicrobial immune response.Part of the PlantExtract TypeExperimental Model TypeBiological EffectMetabolites Isolated or IdentifiedStudy ConclusionsReferencesWholeMethanolic extract*Listeria monocytogenes* C57BL/6J mice infectionIn vivoAntimicrobialImmunomodulatorCpHigher resistance to infection by intracellular pathogens↓ CFU count and severity of lesions in infected mice[47]WholeMethanolic extract*Candida parapsilosis* infection in RAW264.7 cell line andBALB/c miceIn vivo/vitroAntimicrobialImmunomodulatorCpEnhanced PKC-dependent phagocytosis activity and intracellular death (RAW264.7)↑ Resistance to infection in a macrophage-dependent manner (mice).Restriction of *Candida* dissemination(mice)Alleviation of liver and splenic lesions (mice)↑ Phagolysosomal fusion, phagosomal acidification, and lysosomal enzymatic activity of macrophages.[48]Whole Methanolic extract and fractionation*Eimeria tenella* infection in Lohmann chickensMDKB cellsIn vivo/vitroAnticoccidial and immunomodulator(Cp)2*-β-D*-glucopyranosyloxy-1-hydroxytrideca-5,7,9,11-tetrayne2-*β-D-*glucopyranosyloxy-1-hydroxy-5(*E*)-tridecene-7,9,11-triyne3-*β-D*-glucopyranosyloxy-1-hydroxy-6(*E*)-tetradecene-8,10,12-triyneAnticoccidial effects only evident with the extract and Cp↓ Mortality in infected chickens (both)↓ Invasion of sporozoites, interfering with the life cycle of the parasite (Both)↓ Excretion of oocysts in fecal matter↓ Intestinal injury (both)↑ IFN-γ production in LT cells (Extract)[52]FlowersMonofloral honeys from the nectar of *B. pilosa*Bacteria growthinhibition by agar disk-diffusion methodIn vitroAntibacterialPolyphenols and flavonoidsInhibition of the growth of*S. aureus*, *S. intermedius* B, *S. xylosus*, *C. koseri*, hemolytic *E. coli*, and *S. cholearasuis*[54]LeavesHexaneDichloromethaneEthyl acetateAcetoneMethanolMICIn vitroAntibacterialGroup of phenolic compoundsBacterial (*Klebsiella pneumoniae*, commercial probiotics, *E. coli*, *Salmonella typhimurium*, *Shigella boydii*, and *Vibrio parahaemolyticus*) growth inhibition by the dichloromethane fraction [55]Not specifiedEssential oilHuman gingival fibroblastsIn vitroAntimicrobial and Pro-inflammatory-↑ PGE-2 secretion in synergy with IL-1βNo inhibition of IL-6 and IL-8 synthesis [56,58]Abbreviations: (CFU) Colony forming unit; (Cp) Cytopyloine; (IL) Interleukin; (IFN-γ) Interferon-gamma; (MDBK) Madin-Darby Bovine Kidney Cells; (MIC) Minimum Inhibitory Concentration; (PGE2) Prostaglandin E2; (PKC) Protein kinase C; ↑ Increase; ↓ Decrease.


### 3.3. Extracts of B. pilosa Have Antiproliferative and Antitumor Effects in Different Experimental Models through the Induction of Apoptosis

Different pathways, or hallmarks used by tumor cells to survive, have been discovered throughout tumorigenesis and cancer development. These involve selective growth and proliferative advantage, altered stress response favoring overall survival, vascularization, invasion and metastasis, metabolic rewiring, an abetting microenvironment, and immune modulation [68]. The characterization of these hallmarks has opened new avenues for therapeutic approaches and the search for new medicines for the management of cancer.

For instance, concerning antiproliferative activity, new therapies have been developed with chemotherapeutic agents and compounds of natural origin aimed at inducing cancer cell death by apoptosis in an immunogenic context that also activates the patient’s immune response against tumor cells [69]. Some of the articles here reviewed have reported the traditional use of hydroalcoholic solutions of *B. pilosa* for treating tumors in countries such as Cuba [70,71,72]. Likewise, in vitro studies have shown that polyacetylenic, flavonoid, and terpene compounds exerted cytotoxic and chemopreventive effects on different cancer models, as shown in Table 4 [72,73].

Kviecinski et al. evaluated the antitumor activity of two extracts obtained from the aerial part of *B. pilosa*, namely a hydroethanolic crude extract (HCE) and a SCF. Both preparations showed concentration-dependent cytotoxic activity on human breast carcinoma cells (MCF-7), with a half maximal inhibitory concentration (IC_50_) of 811 µg/mL and 437 µg/mL, respectively, with SCF being more cytotoxic. Additionally, the antitumor activity of both extracts was evaluated against Ehrlich ascites carcinoma (EAC) in BALB/c mice. An intraperitoneal dose of 100 mg/kg (per day for 9 days) of either extract significantly reduced the ratio of non-viable/viable tumor cells, body weight, and volume of ascites fluid retained by mice [74]. The authors emphasized that SCF led to a more significant reduction in the ascites fluid volume and induced a greater tumor growth inhibition. These results resembled the effect of traditional chemotherapeutic agents such as doxorubicin. However, the overall conclusion was that HCE and SCF increased mice’s mean survival time and life expectancy, highlighting their antitumor potential [74].

A study by Shen et al. evaluated in vitro the cytotoxic activity of a petroleum ether extract (triterpene enriched) of *B. pilosa* on the following tumor cell lines, HepG2 (hepatocellular carcinoma), CNE-2 (nasopharyngeal carcinoma), B16 (murine melanoma), and A549 (adenocarcinomic human alveolar basal epithelial cells). The petroleum ether extract showed cytotoxic activity against all four human cancer cell lines, especially the A549 one (IC_50_ = 49.11 ± 2.72 μg/mL) [75]. Likewise, the in vivo antitumor effect of the *B. pilosa* extract was examined in the A549-xenograft murine model. The extract, administered orally at doses of 90, 180, and 360 mg/kg during 14 days, inhibited the xenograft growth in nude BALB/c mice; the respective inhibition rates were 24.76%, 35.85%, and 53.07%. Moreover, as demonstrated by Western blotting, the extract downregulated Bcl-2 and upregulated Bax and caspase-3 protein expression; that is, the extract triggered cell death through the mitochondria-mediated apoptosis pathway. These findings suggested that *B. pilosa* extract could be considered a potential chemotherapeutic compound against lung cancer that merits to be studied in the future [75].

In our laboratory, *B. pilosa* extracts and fractions of different polarities have shown a low cytotoxic activity on immune cells such as PBMCs and human macrophages (article in preparation). This observation suggests that the low polarity extracts tested might have selective cytotoxic activity on HepG2, CNE-2, B16, and A549 cell lines.

In another study, Jurkat leukemic cells were pretreated with Cp isolated from the butanolic fraction of *B. pilosa* to evaluate the changes in their proteomic profile. Cp upregulated the expression of 12 proteins involved in signal transduction, detoxification, metabolism, energy pathways, and channel transport [76]. Some of the proteins upregulated in the presence of Cp were Rho GDP-dissociation inhibitor 2 (GDIR2), glutathione transferase omega 1 (GSTO1), hemoglobin beta chain (HBB), protein disulfide isomerase A3 (PDIA3), adenosine deaminase (ADA), D-3-phosphoglycerate dehydrogenase (SERA), DJ-1 protein (PARK7), cytoplasmic NADP-dependent isocitrate dehydrogenase (IDHC), mitogen-activated protein kinase kinase 1-interacting protein 1 (LTOR3), LMNB1 protein (LMNB1), glutathione S-transferase P (GSTP1), and anion-selective channel protein 2 (VDAC2). On the other hand, Cp downregulated the expression of the following nine proteins in Jurkat cells: mitochondrial 39S ribosomal protein L39 (RM39), methylcrotonoyl-CoA carboxylase beta chain (MCCB), thioredoxin-like protein 1 (TXNL1), chromosome 3 open reading frame 60 (C3orf60), BH3 interacting domain death agonist (BID), thioredoxin-dependent peroxide reductase, mitochondrial precursor (PRDX3), 40 kDa peptidyl-prolyl cis-trans isomerase (PPID), thioredoxin-like protein 2 (GLRX3), and NADH-ubiquinone oxidoreductase 13 kDa-B subunit (NDUA5) [76]. Additionally, the mitochondrial membrane potential of the Cp-pretreated Jurkat cells was evaluated. Results showed that Cp induced cell apoptosis in a dose-dependent manner (10, 20, and 30 µg/mL) and confirmed that mitochondrial dysfunction and the ensuing cell apoptosis were related to the up- and downregulation of the proteins mentioned above [76].

The study by Liu et al., previously mentioned, also tested in vitro the cytotoxic effect of monofloral honeys produced from the nectar of several plants (*B. pilosa*, *Dimocarpus longan*, *Litchi chinensis*, *Citrus maxima*, and *Aglaia formosana*) on the human colon carcinoma cell line (WiDr). The MTT assay (3-(4,5-dimethylthiazol-2-yl)-2,5-diphenyltetrazolium bromide) showed that none of the honey samples (200 to 1000 μg/mL) exhibited cytotoxic activity. Furthermore, the anti-inflammatory potential of the honeys was evaluated in terms of the levels of IL-8 synthesized by honey-treated WiDr cells; all honeys largely inhibited IL-8 production, except the honey produced from *B. pilosa* [54].

Some pre-clinical studies have demonstrated the antitumor effect of *B. pilosa*. However, it will be essential to carry out additional studies on cytotoxic activity linked to the release of tumor-associated antigens (TAAs) and danger-associated molecular patterns (DAMPs), which constitute a new field of interest in cancer immunotherapy.

Additionally, it is unknown whether *B. pilosa* might have other molecular targets, e.g., targets in the tumor microenvironment that could be modulated, or whether the plant components could synergize with conventional treatments or counteract the drug resistance mechanisms of tumor cells. A comprehensive evaluation of these aspects is essential to developing new immunotherapeutic phytomedicines to treat cancer effectively.

### 3.4. Extracts of the Aerial Part of B. pilosa and Honey Produced from the Plant’s Flowers Have Antioxidant Properties Which Reduce the Harmful Effects Associated with Oxidative Stress, as Demonstrated in Cellular and Animal Models

Oxidative stress is elicited by an imbalance between the production and accumulation of ROS in cells and tissues, which leads to the oxidation of proteins, lipids, and DNA and their subsequent dysfunction [77,78]. In particular, mitochondrial oxidative stress associated with hydrogen peroxide (H_2_O_2_) production has been linked to the development of chronic inflammation, cancer progression [79], diabetes mellitus [80,81,82], and atherosclerosis [83,84], due to the activation of redox-sensitive transcription factors such as hypoxia-inducible factor 1 alpha (HIF-1α) and NF-κB [85,86,87], the production of pro-inflammatory cytokines, and the activation of inflammasomes, which favors tissue injury by necrotic and apoptotic processes [88,89,90]

In this context, it has been established that natural compounds such as polyphenols could be helpful as anti-inflammatory agents, given their antioxidant properties and capacity to inhibit enzymes involved in the eicosanoid synthesis [91]. *B. pilosa* has been reported to have polyphenols and additional flavonoids, which could also exert antioxidant and protective effects against the damage generated by free radicals, as described in Table 5.

Kviecinski et al. evaluated the antioxidant and hepatoprotective activity of the *B. pilosa* HCE and different fractions obtained in chloroform, methanol, and EtOAc from the aerial part of the plant [92]. First, the in vitro free-radical scavenging activity was evaluated by the DPPH (2-diphenyl-1-picryl-hydrazyl-hydrate) radical scavenger method and the capacity to protect against lipid peroxidation. Results were expressed in terms of IC_50_ of phytoproduct required to inhibit the generation rate of radicals or lipid peroxidation by 50%. The HCE and the EtOAc fraction had the highest antioxidant activity (IC_50_ = 14.2–98.0 μg/mL and 4.3–32.3 μg/mL, respectively).

Subsequently, in vivo assays demonstrated that the pretreatment during 10 days of BALB/c mice with *B. pilosa* extracts (HCE or EtOAc fraction at 15 mg/kg), prevented the increase in lipid peroxidation and protein carbonylation in the liver upon exposure to CCl_4_ (hepatotoxic xenobiotic agent). Additionally, the ferric ion-reducing antioxidant power (FRAP) was totally recovered in the presence of EtOAc (to the level of normal experimental controls) whereas CCl_4_ decreased it [92].

These experiments also showed that the HCE and the EtOAc fraction were able to prevent the CCl_4_-induced depletion of reduced glutathione (GSH) and diminish the sera levels of enzymes associated with liver damage, such as aspartate aminotransferase (AST), alanine aminotransferase (ALT), and lactate dehydrogenase (LDH). Additionally, the EtOAc fraction prevented the fragmentation of DNA in mice hepatocytes. The authors attributed these results to the high content of flavonoids, such as quercetin, identified in the EtOAc fraction [92].

On the other hand, the study by Liu et al., previously mentioned, also evaluated the antioxidant potential of extracts of monofloral honeys produced from the nectar of *B. pilosa, Dimocarpus longan*, *Litchi chinensis*, *Citrus maxima*, and *Aglaia formosana* (see the antimicrobial and antitumoral section). *B. pilosa* honey showed a DPPH radical scavenging activity of 84.9% and an inhibitory effect on hydroxyl radical formation of 80 to 90%, higher than those of the other honeys tested. It also displayed a greater reducing power (equivalent to 7.00 ± 0.05 mg/mL of dibutyl hydroxytoluene used as reference). However, *B. pilosa* honey had little inhibitory effect on the formation of superoxide radicals, which was significantly lower (51.0%) than the other honeys. The authors attributed the antioxidant effects of honey from *B. pilosa* to its high content of proteins (1.68 ± 0.03 mg/g), phenolic compounds (822 ± 0.03 mg/g), and flavonoids (124 ± 0.89 mg) [54].

It has been described that during the early development stages of *B. pilosa*, its leaves have a high content of antioxidant agents, such as phenolic compounds, carbohydrates, ascorbic acid, and carotenoids, which through the elimination of ROS excess protect the plant cells from photo-oxidative damage [93,94,95]. This feature would explain the capacity of *B. pilosa* to adapt to hostile and stressful environments and its medicinal potential [96].

Consequently, a study was conducted to compare the antioxidant potential of African green leafy vegetables, such as *Amaranthus hybridus* L. and *B. pilosa* versus exotic leafy green vegetables such as *Lactuca sativa* L., *Brassica oleracea*, and *B. oleracea* var. capitata during their vegetative development. For this purpose, several parameters were evaluated: the plant component content, the total antioxidant capacity by FRAP and DPPH methods, and the activity of superoxide dismutase (SOD), catalase (CAT), and peroxidase (POD) [96]. CAT, SOD, and POD activity increased significantly during the early vegetative stages of *B. pilosa* (81.1 ± 1.6, 162.8 ± 2.70, and 49.8 ± 0.86 U/mg/min, respectively) and in *A. hybridus* L. (84.7 ± 0.62, 151.5 ± 1.89, and 45.1 ± 1.30 U/mg/min, respectively). In contrast, enzyme activity in exotic vegetables only increased at more mature stages of development. However, the authors emphasized that the DPPH test results did not vary significantly among the plants studied. Moreover, the FRAP test data showed similar increasing trends during the development stages of all the vegetables analyzed [96].

In addition to describing the antibacterial activity of dichloromethane extracts of *B. pilosa* and *Dichrostachys cinerea* (see antimicrobial section), Shandukani et al. also investigated the antioxidant effect of vegetable’s phenolic compounds. The ethyl acetate, acetone, and methanol extracts from both plants showed a high antioxidant activity. However, the antioxidant activity of *B. pilosa* was greater in the ethyl acetate extract; meanwhile, for *D. cinerea*, the antioxidant activity increased with the extract polarity [55]. These results were consistent with the total phenolic content determined. Therefore, a positive association existed between the total phenolic content and the antioxidant activity observed. Furthermore, the cytotoxicity studies on the muscle cell line (C2C12) showed that none of the crude extracts derived from those plants displayed significant toxicity; only a slight decrease in cell viability was observed at high concentrations (1000 μg/mL) [55].

A more recent study analyzed the effect of boiling and in vitro-simulated human digestion on the stability and bioactivity of phenolic compounds present in methanolic extracts from *B. pilosa* and *Spinacia oleracea* leaves [97]. Phenolic content, radical scavenging activity, redox potential, and cellular antioxidant activity on mouse fibroblast (L929) and human colon adenocarcinoma cell line (Caco-2) were evaluated. Boiling favored a higher yield of phenolic compounds. In contrast, the phenolic content and the in vitro antioxidant activity were reduced in both extracts after duodenal digestion. In addition, compared with *S. oleracea*, *B. pilosa* extract better protected Caco-2 cells, low-density lipoprotein (LDL), and plasmid DNA against oxidative damage. However, this antioxidant activity was reduced after in vitro digestion, but enough activity remained to protect cells from oxidative damage [97].

The antioxidant properties of *B. pilosa*, mainly attributed to the presence of polyphenols and flavonoids, are of relevance. This characteristic is one of the mechanisms through which the extracts and components derived from the plant exert anti-inflammatory activity. However, more in vivo studies with animal models are required to understand better the properties of the plant [98]. Additionally, it would be interesting to simultaneously evaluate the antioxidant and anti-inflammatory properties of *B. pilosa*, as has been reported for *Cassia* species and *Andrographis paniculata* [99,100]. The antioxidant activity of *B. pilosa* components could help to prevent the development of degenerative diseases associated with high levels of oxidative stress and ageing, such as atherosclerosis, cancer, cognitive impairment, cardiovascular conditions, and Alzheimer’s and Parkinson’s diseases [101].
pharmaceutics-15-01491-t005_Table 5Table 5Scientific evidence on the antioxidant properties of *B. pilosa* L.Part of the PlantExtract TypeExperimental Model TypeBiological EffectMetabolites Isolated or Identified Study ConclusionsReferenceAerialHCEChloroform fractionEtOAcMethanolic fractionDPPHCCl_4_-induced hepatotoxicity in male BALB/c miceIn vivo/vitroAntioxidantQuercetin 3,3′-dimethyl ether 7-*O-β-D*-glycopyranoside↓ Generation of hydroxyl radicals in vitro, especially EtOAc.↓ Lipid peroxidation and protein carbonylation in the liver (EtOAc as HCE).Total recovery of FRAP (EtOAc).Prevention of GSH depletion (EtOAc as HCE).↓ Serum AST, ALT, and LDH (EtOAc as HCE)Prevention of hepatocyte DNA fragmentation (EtOAc)[92]FlowersMonofloral honeys from the nectar of *B. pilosa*DPPHMeasurement of antioxidant capacityIn vitroAntioxidantPolyphenols and flavonoids↑ Radical scavenging activity↓ Hydroxyl radical formation↑ Reducing power[54]LeavesAqueousCharacterization of the chemical compositionMeasurement of antioxidant enzyme activityIn vitroAntioxidantPhenolic compounds, carbohydrates, ascorbic acid, and carotenoidsHigh content of compounds during early vegetative growth stage↑ Enzymatic activity of SOD, CAT, and POD during early vegetative growth stagesIncreasing FRAP in the developmental stages[96]LeavesHexaneDichloromethaneEthyl acetateAcetoneMethanolAntioxidant activity by DPPHC2C12 cell lineIn vitroAntioxidantGroup of phenolic compoundsHigh antioxidant activity (ethyl acetate, acetone, and methanolic fractions)No cytotoxic effect [55]LeavesMethanolic extractMeasurement of antioxidant/radical scavenger activityL929 cell lineCaco-2 cellsIn vitroAntioxidantPhenolic compounds↑ Antioxidant activity↑ Capture of radicalsProtection of Caco-2 cells, LDL, and plasmid DNA against oxidative damage[97]Abbreviations: (ALT) Alanine aminotransferase; (AST) Aspartate aminotransferase; (C2C12) muscle cell line; (Caco-2) Human colon adenocarcinoma; (CAT) Catalase; (CCl4) Carbon tetrachloride; (DNA) Deoxyribonucleic acid (DPPH) 2, 2-diphenyl-1-picrylhydrazyl; (EtOAc) Ethyl acetate fraction; (FRAP) Ferric reducing/antioxidant power; (GSH) Reduced glutathione; (HCE) Hydroethanolic crude extract; (L929) mouse fibroblast; (LDH) Lactate dehydrogenase; (LDL) Low-density lipoprotein; (POD) Peroxidase, (SOD) Super oxide dismutase; ↑ Increase; ↓ Decrease.


### 3.5. Whole Plant Extracts of B. pilosa Exert an Anti-Diabetic Effect by Modulating the Adaptive Immune Response in Murine Models of Type 1 Diabetes Mellitus

Type 1 Diabetes Mellitus (T1DM) is a chronic autoimmune disease characterized by insulin deficiency and consequent hyperglycemia. This illness results from a complex interaction between environmental factors, the genome, the microbiome, the metabolism, and the immune system of each individual [102,103]. Therefore, the development of new therapies has considered the use of immunomodulators, either individually or in combination, to prevent or reverse the immunological issues observed in T1DM [104]; for instance, anti-CD3 antibodies are combined with traditional pharmacological immunosuppressants such as rapamycin [105]. Studies summarized in Table 6 show that extracts or compounds derived from *B. pilosa* are traditionally used as antidiabetic phytomedicines that act as modulators of the immune response.

A study conducted in non-obese diabetic mice (NOD) showed that a butanol fraction (ButF) of *B. pilosa*, administered intraperitoneally at a dose of 10 mg/kg three times per week, could maintain the normal morphology of the pancreatic islets, as evidenced in hematoxylin and eosin staining. The ButF inhibited the β-cell death and the pancreatic tissue infiltration by leukocytes. These data suggested that the plant extract ameliorated the Th1-mediated autoimmune diabetes in NOD mice [106]. To assess whether ButF treatment induced a generalized suppression of adaptive immunity, the authors sensitized BALB/c mice with ovalbumin (OVA), a T-cell-dependent antigen. The ButF extract increased the levels of IgE and Th2 cytokines in serum, the levels of IgE and IL-5 in bronchoalveolar lavage, and the infiltration of the respiratory tract by eosinophils and mast cells. Additionally, human CD4^+^ T cells cultured in vitro under Th1 (PHA, IL-12 and anti-IL-4) or Th2 (PHA, IL-4 and anti-IL-12) polarizing conditions were exposed to the ButF extract. The extract altered the cell differentiation, namely, it inhibited the Th1 but favored the Th2 polarization by activating the transcription factor GATA-3 [106,107].

The subsequent research of these authors evaluated the antiproliferative activity of Cp obtained from the whole plant. Cp significantly suppressed the proliferation of NOD mice splenic CD4^+^ T cells induced by IL-2/Concavalin A or anti-CD3 antibodies (*p* < 0.05 vs. control) in a dose-dependent manner (2, 5, and 10 μg/mL) [108]. Moreover, Cp suppressed the differentiation of NOD, the differentiation to Th2. Additionally, NOD mice treated with Cp at a dose of 25 μg/kg three times per week, showed decreased serum IFN-γ and increased IL-4 levels. None of the Cp-treated mice developed diabetes suggesting that Cp inhibited the CD4^+^ T-cell infiltration of the pancreatic islets and promoted an anti-inflammatory response that preserved the integrity of this organ. Interestingly, the authors noted that Cp did not alter the number of LB or CD8^+^ T cells in the pancreatic lymph nodes but did reduce the number of CD4^+^ T cells [108].

All these data suggest that Cp acts as a modulator of CD4^+^ T cells rather than as an immunosuppressive agent. This feature represents an advantage over conventional treatments that generate adverse effects associated with overall immunosuppression increasing the risk of microbial infection, tumorigenesis, and toxicity [108,109,110].
pharmaceutics-15-01491-t006_Table 6Table 6Scientific evidence on the antidiabetic properties of *B. pilosa*.Part of the PlantExtract TypeExperimental Model TypeBiological EffectMetabolites Isolated or Identified Study ConclusionsReferenceWholeplantButanol fractionNOD miceHuman CD4^+^Th0 cellsJurkat cellsIn vivo/vitroAntidiabeticPolyacetylenes:2-*β-D-*glucopyranosyloxy-1-hydroxy-5(*E*)-tridecene-7,9,11-triyne3-*β-D*-glucopyranosyloxy-1-hydroxy-6(*E*)- tetradecene-8,10,12-triyne↓ Severity and development of autoimmune diabetesMaintenance of normal morphology of pancreatic islets↓ Beta-cell death and leukocyte infiltration (mice)↓ Th1 cytokine synthesis↑ Th2 cytokine synthesis↓ Th1 cell differentiation↑ Th2 cell differentiation↑ GATA-3 but not T-bet transcription↑ Airway inflammation in OVA-challenged mice[106]WholeplantMethanolic extract and fractionation with ethyl acetateEL-4Primary T cellsPrimary β cellsNOD miceNOD-SCID miceIn vivo/vitroAntidiabeticAntiproliferativeImmunomodulatorCp:2- *β-D*-glucopyranosyloxy-1-hydroxytrideca-5,7,9,11-tetrayneEffective prevention of diabetes development↓ Proliferation of CD4^+^ T cells↓ Differentiation of Th0 to Th1 cells↓ Serum IFN-γ↑ Differentiation of Th0 to Th2 cells↑ Serum IL-4↑ GATA-3 but not T-bet transcription↑ FasL transcription in pancreatic islet cells↑ Long-term phagocytic cellsNo effect of decreased T-cell proliferation on OVA response[108]Abbreviations: (Cp) Cytopyloine; (EL-4) Mouse T-cell line, (FasL) Fas ligand; (IFN-γ) Interferon-gamma; (IL) Interleukin; (NOD) Non-obese diabetic mice; (OVA) Ovalbumin; ↑ Increase; ↓ Decrease.


## 4. Discussion

Compounds of natural origin with immunomodulatory properties are currently used as therapeutic agents in treating autoimmune diseases, inflammatory disorders, and cancer [111]. The scientific evidence compiled in this review shows that in different disease contexts, and depending on the extract or compound used, *B. pilosa* promotes anti- or pro-inflammatory responses through different mechanisms of action (Figure 2).

In the case of anti-inflammatory therapies, some ongoing clinical trials are focused on mucositis. These studies have tested a *B. pilosa*-derived pharmaceutical product combined with curcuminoids (FITOPROT) and shown its anti-inflammatory and antioxidant effects on cellular and animal models of intestinal mucositis [27,28].

Phase I clinical trials to evaluate toxicity and side effects of FITOPROT have shown that no participant experienced toxicity or systemic or local effects in patients when used topically as a mouthwash at different doses (10 mg/mL of curcuminoids plus 20% *v*/*v* of *B. pilosa* L. or 20 mg/mL of curcuminoids plus 40% *v*/*v* of *B. pilosa*) three times daily, for ten consecutive days. The laboratory and clinical parameters were in normal conditions. Side effects observed were low intensity and temporary mucosa/dental surface pigmentation (*n* = 7) and tooth sensitivity (*n* = 4). No cellular genotoxic effects were observed, nor were altered levels of MPO, MDA, nitric oxide, or production of pro-inflammatory cytokines found. Therefore, it was shown that the use of the FITOPROT product is safe at these proven doses. [112].

Additionally, Phase II clinical trials were developed to evaluate the effectiveness of FITOPROT, focused on treating chemo-radiotherapy-induced mucositis in patients with head and neck cancer. It was determined that the administration of FITOPROT could effectively reduce the severity of the lesions, as evidenced by the lower synthesis of pro-inflammatory IL-8 [113].

It has also been determined that the SCF of *B. pilosa* is effective in modulating oxidative stress and immune responses in an animal model of intestinal inflammation by counteracting the GSH depletion and significantly reducing the synthesis of TNF-α and the NF-κB expression. This effect has been associated with long-chain fatty acids such as linolenic acid [43]. Additional studies have shown that linolenic acid also regulates the immune response due to its protective effect against oxidative stress and inflammation, by restoring GSH levels, and decreasing TNF-α synthesis, colonic iNOS expression, and NF-κB activation. This last effect is fundamental in reducing the production of proinflammatory cytokines and transcription factors [114,115].

It is also known that this SCF (fatty acid enriched) of *B. pilosa* has an antiproliferative effect on the breast cancer MCF-7 cells and in vivo antitumor effect in the EAC model. This latter effect has been related to the presence of polyacetylenic components [74].

Although the cellular and molecular mechanisms by which SCF of *B. pilosa* behaves as a cytotoxic agent are unknown, it has been described that Cp, a polyacetylenic component, is cytotoxic for Jurkat cells due to its activity at the mitochondrial level and the ability to induce apoptosis [76]. This apoptotic effect on tumor cells could result in immunogenic cell death (ICD), characterized by the release of DAMPs such as calreticulin, heat shock proteins (HSPs), and adenosine triphosphate (ATP) [116], which in turn trigger the activation of the antitumor immune response.

Although no scientific evidence supports that Cp induces immunogenic death, it will be crucial to characterize the type of death induced in tumor cells and the type of subsequent modulation of the antitumor immune response [117,118].

It has also been reported that Cp can effectively diminish the metastasis of mouse breast cancer 4T1 cells by decreasing the differentiation and function of myeloid-derived suppressor cells (MDSC) [119]. These cells cause immunosuppression and promote cancer progression by attenuating T-cell activity through the production of arginase 1 (ARG1), inducible nitric oxide synthase (iNOS), transforming growth factor beta (TGFβ), IL10, cyclooxygenase 2 (COX-2), indoleamine 2,3-dioxygenase (IDO), vascular endothelial growth factor (VEGF), and ROS. Currently, the eradication of MDSCs from the tumor microenvironment or their reprogramming into pro-inflammatory cells is one of the most studied anti-tumor immunomodulatory strategies and represents a new approach to immunotherapy [120,121,122].

In this regard, natural compounds such as curcumin, obtained from *Curcuma longa*, are able to modulate MDSCs favoring the antitumor immune response against the 4T1 cells. Curcumin reduces the proportion of intratumoral MDSCs by 18.7% and 13.3% depending on the dose tested (25 and 50 mg/kg, respectively). Parallel to the reduction in MDSC, curcumin can reprogram the MDSCs towards the pro-inflammatory M1 type and concomitantly decrease the immunosuppressive M2 type MDSCs, thus favoring the cytotoxic T-cell response against the tumor [122].

Another interesting aspect of *B. pilosa* polyacetylenes is their capacity to attenuate or potentiate the immune response depending on the context in which their immunomodulatory effect is evaluated. On the one hand, Cp acts as an antidiabetic agent in the T1DM model (NOD mice) by reducing the Th1 inflammatory response [106,108]. Cp can modulate T-cell differentiation; specifically, it upregulates the transcription factor GATA-3 and downregulates T-bet; consequently, the production of Th1 and Th2 cytokines is modified. This anti-inflammatory effect has also been observed in Cp isolated from *Bidens pilosa* Linn. var. *Radiata*, which inhibits the differentiation of Th0 to Th1 cells, decreases the synthesis of IFN-γ in mouse splenocytes, and favors the differentiation towards Th2, thanks to the synthesis of IL-4. This control of inflammation leads to protection of pancreatic tissue and reduced destruction of pancreatic β-cells [123]. On the other hand, Cp enhances the immune response generating an antimicrobial protective effect. For instance, in infection models of *Listeria monocytogenes*, *Candida parapsilosis*, and *Eimeria tenella*, Cp can improve the phagocytic activity of macrophages and enhance the production of IFN-γ by T cells [47,48,52].

Despite the lack of additional articles on the pro-inflammatory effect of Cp, it is known that other compounds, such as the flavonoids centaurein and centaureidin from *B. pilosa* var. *Radiata*, also stimulate IFN-γ production by activating the nuclear factor of activated T cells (NFAT) and NF-κB enhancers in Jurkat cells [124]. This mechanism could be associated with the protection of mice against *Listeria monocytogenes* infection and the treatment with centaurein [125].

## 5. Conclusions

There is a large amount of scientific evidence showing that extracts and isolated components of *B. pilosa* have immunomodulatory properties. The anti-inflammatory activity of the plant relates to different chemical components with a broad spectrum of polarity and structural diversity, such as fatty acids and phenolic components. For example, terpenes and quercetin that modulate pain and inflammation are attractive to the therapeutic field for managing autoimmunity, whose pathophysiology is linked to an exacerbated immune response (Table 7). Other studies have described that polyacetylenic components such as Cp could have a dual effect on the immune response by acting as anti- or pro-inflammatory agents, depending on the disease context in which they are studied. For instance, in the T1DM context, Cp can attenuate the proinflammatory Th1 response and favors the Th2 regulatory response. In contrast, in infectious diseases, Cp induces an antimicrobial immune response by promoting phagocytic activity, phagolysosomal fusion, phagosomal acidification, and lysosomal enzymatic activity of infected macrophages; thus, Cp could be an alternative strategy for the management of infections, especially those that do not respond to conventional medicines. Additionally, Cp could also exert immunomodulatory effects potentially useful in the field of antitumor therapy, although this is not entirely clear. The mechanisms by which Cp generate cytotoxicity, induce immunogenic cancer cell death, and modulate the activation of the immune system cells involved in the antitumor response remain to be investigated. The knowledge achieved at the pre-clinical level about the anti-inflammatory properties of *B. pilosa* is a fundamental support for the design of new phytomedicines focused on treating diseases in which there is an imbalance of the immune response.

## Figures and Tables

**Figure 2 pharmaceutics-15-01491-f002:**
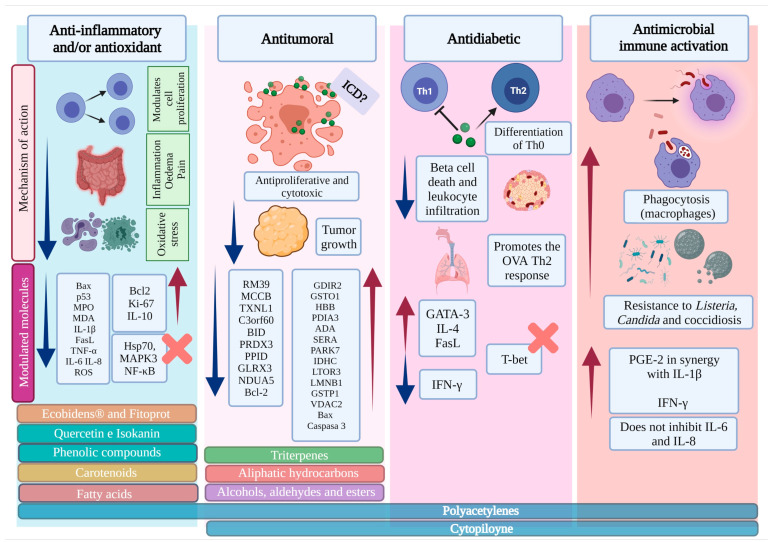
Immunomodulatory activity of *B. pilosa* L. Each section corresponds to the effect induced by the immunomodulatory properties of *B. pilosa* (anti-inflammatory and antioxidant, antitumoral, antidiabetic, and immune antimicrobial activation). In each one of them, the mechanisms of action and the molecules that are modulated by the products and compounds derived from this plant are shown (horizontal boxes). Created with BioRender.com.

**Table 1 pharmaceutics-15-01491-t001:** Terms used in the search.

Item 1	Item 2	Item 3
*Bidens pilosa OR* * *bidens*	AND *	Immunomodulation
		Inflammation
		Anti-inflammatory agents
		Antitumor agents
		Antioxidant effects
		Infections

* OR/AND are the Boolean operators used to search the different databases.

**Table 4 pharmaceutics-15-01491-t004:** Scientific evidence on the antitumor properties of *B. pilosa* L.

Part of the Plant	Extract Type	Experimental Model Type	Biological Effect	Metabolites Isolated or Identified	Study Conclusions	Reference
Aerial	Ethanol extract: water (9:1) fractionated SCF	MCF-7 cell line EAC in BALB/c mice In vitro/vivo	Antitumor and Antiproliferative	Polyacetylenes	Cytotoxic activity in a concentration-dependent manner ↓ Body weight, ascites fluid volume, and tumor cells ↑ Non-viable/viable tumor cells ratio ↑ Inhibition of tumor growth ↑ Mean survival time and life expectancy of mice	[74]
Aerial	Petroleum ether	Cell lines: HepG2 A549 CNE-2 B16 A549-xenograft murine model. In vitro/In vivo	Antitumor and Antiproliferative	Triterpenes Aliphatic hydrocarbon	Antiproliferative activity on tumor cell lines Tumor growth (A549) inhibition in mice ↓ Bcl-2 protein expression ↑ Bax and caspase-3 expression	[75]
Whole plant	Ethanolic crude extract fractionated with ethyl acetate and n-butanol	Jurkat cells (clon E6-1, TIB 152) In vitro	Antitumoral	Cp 2-*β*-D-glucopyranosyloxy-1-hydroxytrideca-5,7,9,11-tetrayne	↑ Expression of the GDIR2, GSTO1, HBB, PDIA3, ADA, SERA, PARK7, IDHC, LTOR3, LMNB1, GSTP1, and VDAC2 proteins ↓ RM39, MCCB, TXNL1, C3orf60, BID, PRDX3, PPID, GLRX3, and NDUA5 protein expression	[76]
Flowers	Monofloral honeys from the nectar of *B. pilosa*	WiDr cell line MTT In vitro	No anti-inflammatory effect No cytotoxic effect	Polyphenols and flavonoids	No cytotoxic effect No inhibition of IL-8 synthesis	[54]

Abbreviations: (A549) Adenocarcinomic human alveolar basal epithelial cells; (ADA) Adenosine deaminase; (B16) murine melanoma; (Bax) BCL-2-like protein 4; (BID) BH3 interacting domain death agonist; (C3orf60) Chromosome 3 open reading frame 60; (Cp) Cytopyloine; (CNE-2) nasopharyngeal carcinoma; (EAC) Ehrlich ascites carcinoma; (GDIR2) Rho GDP-dissociation inhibitor 2; (GLRX3) Thioredoxin-like protein 2; (GSTO1) Glutathione transferase omega 1; (GSTP1) Glutathione S-transferase P; (HBB) Hemoglobin beta chain; (HepG2) Liver hepatocellular carcinoma; (IDHC) Cytoplasmic NADP-dependent isocitrate dehydrogenase; (LMNB1) LMNB1 protein; (LTOR3) Mitogen-activated protein kinase, kinase 1 interacting protein 1; (MCCB) Methylcrotonoyl-CoA carboxylase beta chain; (MCF-7) Human breast carcinoma cells; (MTT) (3-(4,5-dimethylthiazol-2-yl)-2,5-diphenyltetrazolium bromide) (NDUA5) NADH-ubiquinone oxidoreductase 13 kDa-B subunit; (PARK7) Protein deglycase DJ-1; (PDIA3) Protein disulfide isomerase A3; (PPID) 40 kDa peptidyl-prolyl cis-trans isomerase; (PRDX3)Thioredoxin-dependent peroxide reductase; (RM39) Mitochondrial 39S ribosomal protein L39; (SCF) Supercritical fluid; (SERA) D-3-phosphoglycerate dehydrogenase; (TXNL1) Thioredoxin-like protein 1; (VDAC2) Anion-selective channel protein 2; (WiDr) Human colon carcinoma cell line; ↑ Increase; ↓ Decrease.

**Table 7 pharmaceutics-15-01491-t007:** Active components identified or isolated from *B. pilosa* L. associated with its immunomodulatory properties. (Structures taken from Bartolome 2013).

Compound	Type	Chemical Structure	Immunomodulatory Property
2-*β*-*D*-Glucopyranosyloxy-1-hydroxytrideca-5,7,9,11-tetrayne	Polyyne	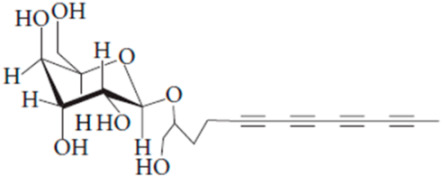	Antimicrobial Antidiabetic Antitumor
2-*β-D*-glucopyranosyloxy-1-hydroxy-5(*E*)-tridecene-7,9,11-triyne	Polyyne	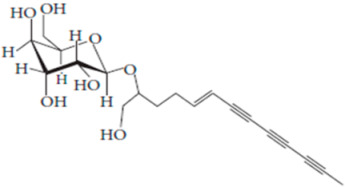	Antimicrobial Antidiabetic
3-*β-D*-glucopyranosyloxy-1-hydroxy-6(*E*)-tetradecene-8,10,12-triyne	Polyyne	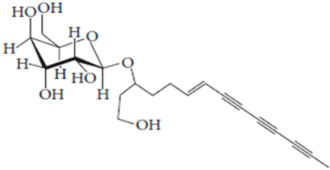	Antimicrobial Antidiabetic
2*-O*-*D*-glucosyltrideca-11*E*-en-3,5,7,9-tetrayn-1,2-diol	Polyyne	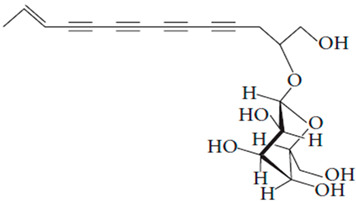	Anti-inflammatory
Quercetin 3,3′-dimethyl ether 7-*O-β-D*-glucopyranoside	Flavonoid	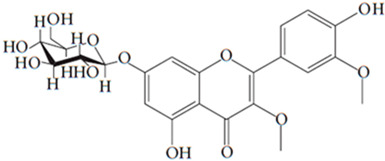	Anti-inflammatory Antioxidant
Iso-okanin 7*-O-β-D*-(2″,4″,6″-triacetyl)- glycopyranoside	Flavonoid	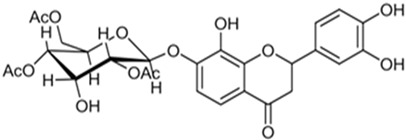	Anti-inflammatory
Quercetin-3*-O-*robinobioside	Flavonoid	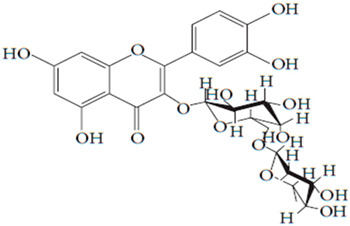	Antioxidant
Rutin	Flavonoid	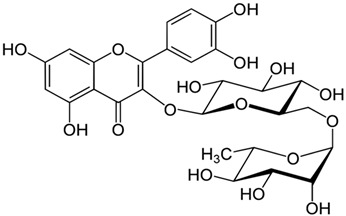	Antioxidant
Quercetin-3-*O*-glucoside	Flavonoid	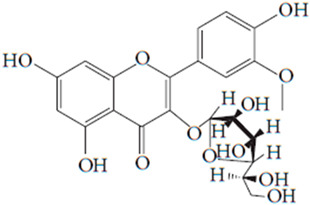	Antioxidant

## Data Availability

Not applicable.

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
