# Peer review of "Immunomodulatory Properties of Natural Extracts and Compounds Derived from Bidens pilosa L.: Literature Review"

_pharmaceutics, 2023, doi:10.3390/pharmaceutics15051491_

Round 1

Reviewer 1 Report

The submitted manuscript presents the review o papers concerning Bidens pilosa. The description of the screening of available literature is very detailed. However, there is a kind of discrepancy between the list of references (122) and the statement “Finally, 23 articles 108 were included in the present review.”

Due to the many reviews concerning this plant, it is suggested to highlight the novelty of this review.

Please use mice instead of “Mus musculus” like it is in the case of “rats” in the main text.

Pease remove “(Rattus norvegicus albinus)” from the table.

L. 407 “1000 to 200 μg/ml”, it is suggested “200 to 1000 μg/ml”

The editorial changes must be provided to improve the quality of the presentation in tables. Please check the lines in the tables.

Please check abbreviations because there is no need to repeat the full names, e.g. DAMPs and IFN.

Author Response

Consulte el archivo adjunto.

Reviewer 2 Report

This review presents the evidence regarding the ability of Bidens pilosa to exert anti-inflammatory, antioxidant, analgesic, immunomodulatory, antitumor , antioxidant, antidiabetic effects. The review is comprehensive and well organized. The tables are well structured and informative. The readability of the article might be increased by one or more figures pointing out the main mechanisms on which the plant acts.

Reviewer 3 Report

I have enjoyed reading the manuscript entitled: Immunomodulatory properties of natural extracts and compounds derived from Bidens pilosa L.: Literature review by Santander González et al.

In general, the article is easy to read, quite well designed and can be of interest to readers and researchers. However, further details and some suggestions on how to improve your work are described below:

-       Please define more precisely the inclusion and exclusion criteria used in the material and methods section.

-       It would be interesting if these biological effects of B. pilosa would be presented in subchapters: in vitro, in vivo and clinical trials. At the same time, the mechanisms of action underlying these biological effects could be described in more detail

-       Please clarify the copyright issue about tables 2, 3, 4, 5 and 6.

-       Please check for double spacing. English language, although reasonably good, should be improved. This will apply to the whole manuscript.

-        Please summarize the main theme of the review in a graphical abstract.

The order of the authors differs in the article compared to the platform, PLEASE CLARIFY.

Reviewer 4 Report

The present article written by Rodriguez-Mesa XM et al. presents important information regarding Bidens pilosa L.

However, here are some suggestions of improvement:

-        Line 34: I would suggest to add an acronym for Bidens Pilosa and later to use it through the whole text

-        Line 35: Please add other common names for the plant, not just those from Latin America

-        Please see the journal’s instructions for authors regarding the references writing.

-        The entire document needs English editing

-        Line 92: Please add a reference for Figure 1

-        Please describe the acronyms the first time you use them in the text. (e.g.: Line 98, 99) Moreover, the abbreviation list needs to be alphabetically ordered and rescanned, as some abbreviations are missing.

-        I would suggest to add a table/paragraph regarding the phytochemical composition of the plant, so that the reader can better understand its composition

-        Line 129: Please explain the peripheral mechanism of acetylsalicylic acid.

-        There are many mechanisms stipulated throughout the whole manuscript, without a short description (so that the reader can better understand its impact/significance)

-        Line 142: “oral administration, .... mainly at a dose of 100mg/kg” … how long? Please include the period. Please do the same for the entire manuscript

-        Line 159: MPO ? MDA ?

-        Discussion part: you described some ongoing clinical trials. In the abstract you’ve only mentioned the pre-clinical studies. Please revise the abstract accordingly.

-        Moreover, please include more information regarding the side effects observed/toxicity and also the treatment duration.

Reviewer 5 Report

The manuscript "Immunomodulatory properties of natural extracts and compounds derived from Bidens pilosa L.: Literature review" is very complex and interesting and could be published.

- What do the items in Table 1 represent: OR and items 2 AND?

- If there are immunostimulant formulations and how are they administered?

Round 2

Reviewer 3 Report

After the revision, the manuscript is greatly improved.

Reviewer 4 Report

The authors addressed all my comments. Thank you